

# Magnetic signatures of natural and anthropogenic sources of urban dust aerosol

Haijiao Liu[1,2], Yan Yan[3], Hong Chang[1], Hongyun Chen[4], Lianji Liang[5], Xingxing Liu[1], Xiaoke Qiang[1], Youbin Sun[1,6]

[1]State Key Laboratory of Loess and Quaternary Geology, Institute of Earth Environment, Chinese Academy of Sciences, Xi'an, 710061, China

[2]College of Earth Science, University of Chinese Academy of Sciences, Beijing, 100049, China

[3]Guangzhou Institute of Geochemistry, Chinese Academy of Sciences, Guangzhou, 510640, China

[4]Research Center for Loess and Global Changes, Institute of Hydrogeology and Environmental Geology, Chinese Academy of Geological Sciences, Shijiazhuang, 050803, China

[5]College of Architecture and Civil Engineering, Beijing University of Technology, Beijing, 100124, China

[6]Institute of Global Environmental Change, Xi'an Jiaotong University, Xi'an, 710049, China

Corresponding to: Youbin Sun (sunyb@ieecas.cn)

**Abstract.**The characteristics of urban dust aerosols and the contributions of their natural and anthropogenic sources are of scientific interest as well as being of substantial sociopolitical and economic concern. Here we present the results of a comprehensive study of dust flux and magnetic signatures, including magnetic susceptibility ($\chi$) and the morphology and elemental composition of magnetic particulates, of atmospheric dustfall originating from natural dust sources in East Asia and local anthropogenic sources in Xi'an, China. The results reveal a significant inverse relationship, on a seasonal basis,

between variations in dust flux and $\chi$. By comparing $\chi$ records of desert surface sediments and local polluted dust, the relative contributions of natural and anthropogenic sources can be estimated for the urban atmospheric dustfall. Analysis using Scanning Electron Microscopy (SEM) combined with Energy Dispersive Spectroscopy (EDS) indicates that magnetic particulates from different sources have distinctive morphological and elemental characteristics. Detrital magnetic particles originating from natural sources are characterized by relatively smooth surfaces with Fe and O as the major elements and a

minor contribution from Ti. The anthropogenic particles have angular, spherule, aggregate, and porous shapes with distinctive contributions from marker elements, including S, Cr, Cu, Zn, Ni, Mn and Ca. Our results demonstrate that this



multidisciplinary approach is effective in distinguishing dust derived from distant natural sources and local anthropogenic sources, and for quantitative assessment of the contributions of the two end-members.

## 1 Introduction

Urban dust aerosols, comprising both natural and anthropogenic contributions with complex morphological and physiochemical characteristics, have become a focus of studies of global climate change and regional air pollution (Wilson et al., 2002). Natural dust is derived primarily from long-range transport, with a minor local soil contribution, and causes dust events including sandstorms, suspended-dust and blown-sand weather (Sun et al., 2001; Zhang et al., 2003a; Chen et al., 2004; Kan et al., 2007; Baddock et al., 2013), and has a deleterious effect on local air quality (Wang et al., 2004; Ginoux et al., 2004). Anthropogenic dust produced by human activities is characterized by high concentrations of toxic heavy metals (e.g., Pb, Zn, Co, Cr, Ni, As), which has a long-lasting and more adverse impact on local environment and human health (Zdanovicz et al., 2006; Qiao et al., 2013; Lu et al., 2014; Lee et al., 2015).

Airborne particulate matter ＜2.5 μm in diameter (e.g., PM2.5 and PM1) can enter the alveolar region and even blood circulatory system leading to health issues and even death (Brunekreef, et al., 2002; Nel et al., 2006; Pickrell et al., 2009; Maher et al., 2013). Moreover, anthropogenic dust as an important medium for the formation of secondary pollutants and plays a significant role in the formation of haze events (Hanisch and Crowley, 2001; Li et al., 2001; Lee et al., 2002; Usher et al., 2002; Finlayson-Pitts et al., 2003; Rubasinghege and Grassian, 2009; Takeuchi et al., 2010; Wu et al., 2011; Huang et al., 2014). Consequently, it is important to distinguish the characteristics and contributions of natural and anthropogenic dusts in urban aerosols to formulate effective policies for city administrations on abating dust pollution and improving environmental quality.

Natural and anthropogenic contributions to urban dust aerosols are usually assessed quantitatively using geochemical and magnetic methodologies (Gorden, 1988; Xie et al., 1999; Gomez et al., 2004; Spassov et al., 2004; Kim et al., 2009; Feng et al., 2012). Geochemical methods typically involve source apportionment and contribution assessment of representative heavy metal elements using statistical methods such as chemical mass balance (CMB) (Chow et al., 2002; Gupta et al., 2006)



and factor analysis (FA) (Harrison et al., 1997a; Salvador et al., 2004). According to their sensitivity to various anthropogenic factors, Pb, Fe, Zn, Cr, Cd, Ni, Ba and Sb are frequently used as marker elements for vehicle emissions (Huang et al., 1994; Adachi et al., 2004; Meza-Figueroa et al., 2007), while Hg, Pb, Mn, Cr, Co, Cu, Cd and Ni are indicative of coal-combustion sources (Vouk and Piver, 1983; Pacyna and Pacyna , 2001; Sushil and Batra., 2006).

Since magnetic measurements are rapid, inexpensive and non-destructive, environmental magnetism is increasingly being used as an effective approach to study urban dust pollution (Hoffmann et al., 1999; Maher et al., 1988, 1998). By combining magnetic properties with morphological (Muxworthy et al., 2001; Urbat et al., 2004; Blaha et al., 2008a), heavy metal (Hunt et al., 1984; de Miguel et al., 1997; Blaha et al., 2008b; Maher et al., 2008), and back trajectory characteristics (Li et al., 2009; Wehner et al., 2008; Fleming et al., 2012), the provenance, transport routes and spatial distribution of polluted dust

aerosols can be investigated. This multi-disciplinary approach is becoming a popular means of urban pollution monitoring and assessment (Jordanova et al., 2014; Stein et al., 2015; Yan et al., 2015; Bourliva et al., 2016).

     Using environmental magnetic techniques to assess pollution levels and sources, different forms of urban dust aerosols in East Asia have been studied, including atmospheric dustfall, street dust, leaf dust, inhalable particulate matter, and surface soil. For example, spatial and temporal pollution patterns can be quantitatively estimated from seasonal fluctuations in the

15 concentration and grain-size of magnetic particles in urban roadside dust (Kim et al., 2007, 2009). A high correlation between magnetic parameters ($\chi$ and saturation remanence, SIRM) and heavy metal concentrations in street dust, polluted farmland soil and atmospheric dustfall indicates that these magnetic parameters can be employed as effective proxies for assessing heavy metal pollution (Zhang et al., 2011, 2012a, b; Qiao et al., 2013). The SIRM of the leaves of roadside trees can reflect spatial variations of magnetic particles in urban environments (Hansard et al., 2011, 2012; Quayle et al., 2010;

Maher et al., 2013; Kardel et al., 2012). Though previous studies have investigated the morphology, grain-size, and mineral and elemental composition of urban dust aerosols, there is a lack of studies which use a source-to-sink comparison between magnetic signatures in natural dust, urban dust aerosol, and polluted dust.

     In this study, we systematically collected surface sediments from potential dust sources in East Asia, and typical urban dust aerosols in Xi'an, including consecutive atmospheric dustfall over five years, urban street dust, and typical



anthropogenic pollutants such as vehicle exhausts and fly ashes. Morphology and elemental compositions of magnetic particles in representative samples were analyzed to achieve a thorough source-sink comparison. Our results indicate that natural and anthropogenic contributions to urban dust aerosols can be differentiated using a combination of their magnetic, morphological and elemental characteristics.

## 2 Sampling and methods

### 2.1 Sampling

Natural surface sediments were collected in potential dust sources of East Asia, including the northern Chinese deserts (NCD, the Badain Juran and Tengger Deserts), the Taklimakan Desert (TD), the Mongolian Gobi (MG) and the Tibetan Plateau (TP) (Fig. 1a). Fine-grained materials were taken from alluvial fans, dry riverbeds, lake basins, and drainage depressions within Gobi/sandy deserts at intervals of 100 to 200 km (Fig. 2a–d). To better understand the different sedimentary characteristics, 48 samples from the NCD, 50 samples from the TD, 23 samples from the MG, and 32 samples from the TP were selected for magnetic susceptibility measurement of bulk samples, and morphological and elemental measurements of the magnetic particles in these samples. Locations of the samples are shown in Fig. 1 and descriptions are given in Sun et al., (2013).

Sixty-eight street dust samples were collected from parks, construction sites, commercial streets, and residential areas in Xi'an following a 3×4 km grid spanning approximately 30 km from west to east, and 20 km from north to south (Fig. 1b). The sampling grid covers a range of different functional areas in Xi'an, including Industrial District, Commercial District, Cultural District, Ecological District, and Han Chang'an city ruins park (Fig. 1b). We also collected four typical anthropogenic pollutant samples, including one sample of vehicle exhausts from the exhaust pipes of several vehicles, one sample of fly ashes from electrostatic precipitators at the Baqiao thermal power plant, one street dust sample from the Bell Tower in downtown Xi'an which experiences daily traffic jams, and one street dust sample from near Baqiao thermal power plant where coal-burning is the leading pollution factor. The locations of these samples are shown in Fig. 1.

Atmospheric dustfall collectors were placed on the top of a four-story building at the Institute of Earth Environment, Chinese Academy of Sciences, ~10 m above the ground surface, and a 15-story building inside the Xinxinjiayuan residential





community, ~50 m above the ground surface (Fig. 2e–f). The sampling sites situated in southwest Xi'an consist primarily of commercial and residential districts. Samples were collected using the wet-collection method (Qian and Dong, 2004) at time intervals of 3–5 days in spring and 6–7 days in other seasons. Detailed sampling procedures were reported by Yan et al. (2015a, b). 733 samples were collected from March 2009 to March 2014. Dust flux (DF), g m$^{-2}$ day$^{-1}$ is calculated as follows:

DF=W/ (A×T)

where W is the sample weight in g, A is the area in m$^2$, and T is time in days.

## 2.2 Methods

The magnetic susceptibility of all the bulk samples was measured to estimate the concentration of magnetic minerals. Low- and high-frequency magnetic susceptibilities were measured using a Kappabridge MFK1-FA at frequencies of 976 Hz and

15,616 Hz, respectively. Frequency-dependent susceptibility was calculated as follows:

$\chi_{fd}\%=(\chi_{lf}-\chi_{hf})/\chi_{lf}\times100$

where $\chi_{fd}$ is the percentage frequency-dependent susceptibility, and $\chi_{lf}$ and $\chi_{hf}$ are the low- and high-frequency susceptibilities, respectively, both in m$^3$ kg$^{-1}$. Subsequently, we selected 5 samples from each natural dust source with modal $\chi_{lf}$ values, 4 dustfall samples with high $\chi_{lf}$ and low $\chi_{lf}$, 2 street dust samples with high $\chi_{lf}$, and the 2 samples of vehicle exhausts and fly

ashes as representative samples. The magnetic components of these samples were separated from the bulk samples using a 1 T magnet sealed in a polyethylene bag.

Direct observations of the representative samples and their extracted magnetic particles to confirm their mineral species and morphological and elemental characteristics were performed using a ZEISS EVO-18 Scanning Electron Microscopy (SEM), equipped with Bruker XFlash 6130 Energy Dispersive Spectroscopy (EDS). The specified resolution of the SEM

was <5 nm. The EDS detector is capable of detecting elements with atomic numbers ≥5 and the detection sensitivity can reach 0.1 wt%. Bulk samples and magnetic extracts were characterized by randomly selecting 3–4 fields of view and examining all the particles observed within the selected fields. All the measurements were made at the Institute of Earth Environment, Chinese Academy of Sciences, Xi'an.



## 3 Results

### 3.1 Spatial variation of χlf and χfd in natural dust sources

The $\chi_{lf}$ values of the surface sediments varies from 7.1–88.9×10$^{-8}$ m$^3$ kg$^{-1}$ (Fig. 3a), while their $\chi_{fd}$ values range from

0.4–11.5 % (Fig. 3b). Both $\chi_{lf}$ and $\chi_{fd}$ exhibit a distinctive distribution pattern in different sources. In the TD, $\chi_{lf}$ varies from

12.5–40.3×10$^{-8}$ m$^3$ kg$^{-1}$, with a unimodal distribution peaking at around 20–30×10$^{-8}$ m$^3$ kg$^{-1}$ (Fig. 3a), while $\chi_{fd}$ ranges from

3.0–11.5 % (Fig. 3b). $\chi_{lf}$ in the NCD is also unimodally distributed, ranging from 12.5–40.3×10$^{-8}$ m$^3$ kg$^{-1}$ with peak values at

around 30–40×10$^{-8}$ m$^3$ kg$^{-1}$ (Fig. 3a), while $\chi_{fd}$ varies from 0.4–7.2% (Fig. 3b). In the MG, $\chi_{lf}$ ranges from 19–72.4×10$^{-8}$ m$^3$

kg$^{-1}$, with a bimodal distribution peaking at around 30–40×10$^{-8}$ m$^3$ kg$^{-1}$ and 50–60×10$^{-8}$ m$^3$ kg$^{-1}$ (Fig. 3a), while $\chi_{fd}$ varies

from 1.8–7.5 % (Fig. 3b). In the TP, $\chi_{lf}$ has a multimodal distribution in the range of 7.1–88.9×10$^{-8}$ m$^3$ kg$^{-1}$ with the highest

peak at around 10 and 20×10$^{-8}$ m$^3$ kg$^{-1}$ (Fig. 3a), while $\chi_{fd}$ varies from 0.7–8.9 % (Fig. 3b). The different distribution patterns

of $\chi_{lf}$ indicate that the magnetic minerals in the NCD and TD are different from those in the MG and TP.

The average $\chi_{lf}$ and $\chi_{fd}$ values of natural surface sediments are 32.9×10$^{-8}$ m$^3$ kg$^{-1}$ and 4.8 %, respectively. Average $\chi_{lf}$ in

individual sources shows a decreasing trend from the MG (46.8×10$^{-8}$ m$^3$ kg$^{-1}$), to NCD (38.4×10$^{-8}$ m$^3$ kg$^{-1}$) and TP

(29.6×10$^{-8}$m$^3$ kg$^{-1}$), and then to TD (23.6×10$^{-8}$ m$^3$ kg$^{-1}$). The mean values of $\chi_{fd}$ in different natural sources show a

decreasing trend from the TD (6.9 %) to MG (5.1 %) and TP (4.6 %), and then to NCD (2.5 %).

### 3.2 Spatial variation of χlf and χfd in urban dust samples

The $\chi_{lf}$ and $\chi_{fd}$ values of the urban dust samples, including street dust and atmospheric dustfall, vary from 90.4–1080.7×10$^{-8}$

m$^3$ kg$^{-1}$ (Fig. 3c), and from 0.8–10 % (Fig. 3d), respectively. However, the $\chi_{lf}$ values of the street dust (239.5–1080.7×10$^{-8}$ m$^3$

kg$^{-1}$, mean 524.8×10$^{-8}$ m$^3$ kg$^{-1}$) are higher than those of the atmospheric dustfall (90.4–972.2×10$^{-8}$ m$^3$ kg$^{-1}$, mean 390.8×10$^{-8}$

m$^3$ kg$^{-1}$). $\chi_{fd}$ ranges from 3.4–10.0 % (mean 6.1 %) for the street dust (Fig. 3d), and from 0.8–9.0 % (mean 5.4 %) for the

atmospheric dustfall (Fig. 3d). Low $\chi_{lf}$ (<500×10$^{-8}$ m$^3$ kg$^{-1}$) occurs in the Ecological District, Han Chang'an city ruins Park,

and Cultural District, while samples with intermediate $\chi_{lf}$ values (500–800×10$^{-8}$ m$^3$ kg$^{-1}$) are from the moderately developed

Industrial District and the periphery of the Commercial District. In contrast, the central areas of the Industrial District and the



Commercial District (particularly the area of high traffic density at the Bell Tower) are characterized by relatively high $\chi_{lf}$ values ($>800\times10^{-8}$ m$^3$ kg$^{-1}$). The frequency distributions of $\chi_{lf}$ for the street dust and atmospheric dustfall are unimodal with peaks at around 500–600 and 300–400$\times10^{-8}$ m$^3$ kg$^{-1}$, respectively (Fig. 3c).

Both $\chi_{lf}$ and dust flux of atmospheric dustfall from XA1 and XA2 exhibit significant and consistent seasonal variations (Fig. 4), which suggests that they can represent the dustfall characteristics of Xi'an on a large spatial scale. The lowest (highest) $\chi_{lf}$ values correspond to highest (lowest) dust flux in spring (autumn), which suggests that both natural and anthropogenic contributions to urban dust aerosols have a distinct seasonal pattern.

The representative anthropogenic pollutants, i.e. the vehicle exhausts, fly ashes, and nearby street dust at the Bell Tower and thermal power plant, have high $\chi_{lf}$ (537.9–925.7$\times10^{-8}$ m$^3$ kg$^{-1}$) and $\chi_{fd}$ (8.5–11.1 %). The $\chi_{lf}$ and $\chi_{fd}$ of vehicle exhausts (925.7$\times10^{-8}$ m$^3$ kg$^{-1}$, 11.1 %) and fly ashes (770.0$\times10^{-8}$ m$^3$ kg$^{-1}$, 9.4 %) are higher than the mean values of street dust (524.8$\times10^{-8}$ m$^3$ kg$^{-1}$, 6.1 %) and atmospheric dustfall (390.8$\times10^{-8}$ m$^3$ kg$^{-1}$, 5.4 %).

### 3.3 Morphology and mineralogy of the dust samples

To determine the mineralogical characteristics of the representative bulk samples, more than 40 images were obtained randomly across the sample on the double-sided carbon tape mounted on a SEM stub. For comparison, images were obtained for various types of particles at the same magnification. The morphologies and mineral compositions of representative bulk samples of the natural surface sediments, street dust, and atmospheric dustfall with low and high $\chi_{lf}$ are illustrated in Fig. 5. The particles are typically angular and irregularly shaped in the surface sediments, with a broad size range (around 1–100 μm). Based on the EDS analysis for each particles of the selected field, clay minerals, quartz, calcite, dolomite and magnetic grains (Fig. 5a) are clearly identified (Welton et al., 2012).

The SEM-DES analysis shows that the morphology and composition of the particles in the street dust are complex and heterogeneous. Three categories of particles can be morphologically differentiated, including irregular and aggregate mineral particles, spherical particles, and anomalous particles with poriferous and loose structure (Fig. 5b). Particles with irregular shapes are mainly minerals and commonly present in street dust samples. Compared to the natural surface sediments, the grain-size of mineral particles in the street dust is finer, and mostly ranging from 1–50 μm, with some up to 80 μm. Spherical

particles are mainly amorphous silicon-aluminum and iron-rich spheres, whose grain-size varies mostly from 1–20 μm with some up to 50 μm. There are a small number of anomalous particles with diameters of 10–100 μm.

The morphology and mineral composition of atmospheric dustfall are similar to those of the street dust, except that atmospheric dustfall with low $\chi_{lf}$ has a higher content of irregularly-shaped detrital minerals (Fig. 5c), while that with high $\chi_{lf}$ contains more spheres and anomalous particles (Fig. 5d).

**3.4 Elemental compositions of magnetic minerals**

Since the elemental compositions of mineral particles can be clearly distinguished using SEM-EDS analysis (Blanco et al., 2003; Barbara et al., 2006), a street dust sample dominated by anthropogenic inputs, which has the highest $\chi_{lf}$ in the street dust samples, was selected for analysis. A field of view is shown in Fig. 6. The various mineral particles exhibit distinct chemical compositions. The platy aggregates (labeled a) with high levels of Si and Al, and low levels of K, Ca Mg and Fe are clay minerals composed of crystalline sheet-structure silicates, with a small particle size (Fig. 6a). The angular and sharp-edged particle (labeled b) with high Si and O is quartz (Fig. 6b). The angular particle consisting of Si, Al, and K is potassium feldspar (Fig. 6c). Angular particles with the high levels of Ca and Mg are calcite (Fig. 6d) and dolomite (Fig. 6e).

The angular particles (labeled f) which are abundant in Fe are identified as magnetic grains (Fig. 6f), although some of the particles show low levels of crustal elements, including Si, Al, Ca, and K. Two types of spheres are observed. One (labeled g) is an amorphous alumino-silicate particle (Fig. 6g) with predominant Si and Al and lesser amounts of K, Mg, Na and Ti. These particles have slight variations in element proportions but they vary in size. The other (labeled h) is an iron-rich sphere (Fig. 6h), which is mainly composed of Fe. These particles exhibit various surface textures. In addition, almost all particles contain O and C.

**4 Discussion**



### 4.1 Contributions of local and polluted sources to urban dust

$\chi_{lf}$ can be used as a qualitative proxy for the concentration of magnetic minerals, which is largely controlled by the concentration of ferromagnetic minerals (Dunlop et al., 1997; Evans et al., 2003; Liu et al., 2012). By contrast, $\chi_{fd}$ is sensitive to the superparamagnetic (SP) component. If $\chi_{fd}$ is < 4 %, the proportion of SP particles is likely to be low, while there is a significant SP component if $\chi_{fd}$ is >4% (Dearing et al. 1996). In comparison, the low $\chi_{lf}$ and $\chi_{fd}$ of natural surface sediments indicate an overall low concentration of magnetic minerals, including SP particles, while urban dust aerosols (street dust and atmospheric dustfall) a higher content of magnetic minerals and a high-proportion SP particles (Fig.7). Typical anthropogenic pollutant samples have the highest $\chi_{lf}$ and $\chi_{fd}$ values, which suggests that the anthropogenic pollutants are enriched in the magnetic materials and especially the SP particles.

On the bivariate-plot of $\chi_{lf}$ vs. $\chi_{fd}$, atmospheric dustfall is intermediate between the surface sediments and street dust, implying that atmospheric dustfall is a mixture of distal natural dust and local anthropogenic dust, but much closer to the latter (Fig. 7b). Atmospheric dustfall can be considered as a mixture of distal and local contributions, as well as a mixture of natural and anthropogenic contributions. We assume that the local contribution (LC) consists of both natural and anthropogenic dust, while the anthropogenic contribution (AC) is mainly from two pollutant sources, i.e. vehicle emissions and fly ashes from thermal power plants. Considering that natural dust comes primarily from natural dust sources with a minor local soil contribution (Wang et al., 2004; Ginoux et al., 2012), we attribute the natural contribution (NC) entirely to the distal natural dust.

To quantify contributions of different end members in atmospheric dustfall, the average $\chi_{lf}$ ($25\times10^{-8}$ $m^3$ $kg^{-1}$) of the surface sediments was used as the representative $\chi_{lf}$ value of the NC end-member, while the average $\chi_{lf}$ of the street dust ($550\times10^{-8}$ $m^3$ $kg^{-1}$) and anthropogenic pollutants ($850\times10^{-8}$ $m^3$ $kg^{-1}$) were taken as representing the LC and AC end-members, respectively. On this basis, we calculated the contributions of the LC and AC to urban dust aerosols using the following equations:

LC=($\chi_m$-25)/ (550-25) ×100%,

AC=($\chi_m$-25)/ (850-25) ×100%





where LC is the percentage of the local contribution, AC is that of the anthropogenic contribution, and $\chi_m$ is the

monthly average $\chi_{lf}$ value in $10^{-8}$ $m^3$ $kg^{-1}$. Note that when $\chi_m$ is larger than the average $\chi_{lf}$ of the SD, LC is taken to be

100%.

Results show that both the LC and AC can represent contributions of anthropogenic sources to atmospheric dustfall,

with high values corresponding to larger contributions from anthropogenic sources. The LC values show a distinctive

seasonal pattern (Fig. 8), with the highest values in autumn (92.4 %) and the lowest values in spring (53.1 %), and

intermediate values in winter and summer (74.7 % and 71 %, respectively). The AC shows a similar seasonal pattern to

the LC, with the maximum in autumn (65.6 %), followed by winter (48.4 %), summer (46.6 %), and spring (33.9 %).

Both LC and AC are the lowest in spring, implying that distant natural dust input makes a major contribution to

atmospheric dustfall in spring. The LC and AC variations exhibit a similar seasonal pattern with $\chi_{lf}$, but they are the

opposite to that of dust flux. This suggests that the major sources of atmospheric dustfall varied seasonally between the

distant natural sources in spring and local anthropogenic sources in other seasons. In spring, dust is emitted from the

natural sources by strong winds, and after long-range transport it contributes to the elevated dust flux in Xi'an, and

decreases the LC and AC in atmospheric dustfall. However, from summer to winter, dust input from local

anthropogenic sources is low and stable, indicated by high LC and AC.

## 4.2 Magnetic characteristics of anthropogenic particles

SEM-EDS analysis shows that the extracted magnetic particles from the street dust and atmospheric dustfall can be divided

into detrital and anthropogenic types (Fig. 9). Detrital particles are angular and characterized by relatively smooth surfaces

(Fig. 9a) with Fe and O as the major elements and minor Ti (Fig. 9d), indicating the presence of magnetite, hematite, and

titanomagnetite (Maher et al.,1991; Liu et al, 2015). Anthropogenic particles include angular particles with coarse surface

textures, spherules, aggregates, and porous particles with complex internal structures. The major elements identified in these

particles are Fe and O, which indicate the occurrence of magnetite or hematite, consistent with previously identified

anthropogenic magnetic particles (Kim et al., 2007; Koukouzas et al., 2007; Maher et al., 2009). Minor concentrations of S,

Zn, Cu and Cr are also observed in this type of particle, which is typically attributed to anthropogenic activities (Fig. 9d).





The relatively weaker signal intensity of Fe in the EDS spectra of porous particles indicates a much lower Fe concentration (minimum less than 10 %), while their concentrations of Si, Al, Ca, Ti and Mn are higher.

SEM-EDS analysis shows that the morphology and concentration of magnetic materials in urban dust aerosols varied with sampling sites and over time. Among more than 20 analyzed magnetic extracts from urban dust samples, angular particles with coarse surface textures were the most frequently observed ($>$ 50 %, some up to 80 %), with a wide range of grain size (1–100 μm). Spherules were also commonly observed in all samples, ranging from 10–40 %, mainly with diameters from 10–30 μm. The aggregates with diameters of 5–30 μm account for less than 10 %. Detrital particles, characterized by smooth surfaces, range from 1–5 % and have small diameters (10–20 μm). Porous particles are the least observed magnetic particles (< 1 %) with diameters of 30–120 μm. The SEM-EDS data show that the morphology and concentration of magnetic particulates in atmospheric dustfall with high $\chi_{lf}$ values are similar to those of the street dust, whereas atmospheric dustfall with low $\chi_{lf}$ contains more angular-subangular magnetic particles of detrital origin.

## 4.3 Potential sources of anthropogenic magnetic particles

Anthropogenic magnetic particles in the urban environment are mainly derived from the combustion of fossil fuels (Flanders, 1994; Matzka and Maher, 1999; Muxworthy et al., 2001), vehicle emissions (Harrison et al., 1997b; Moreno et al., 2003; Diapoui et al., 2008; Pant et al., 2013; Maher et al., 2013), and industrial activities (Hanesch et al., 2003; Desenfant et al., 2004). To clarify potential sources, microscopic and elemental investigations of magnetic extracts from anthropogenic pollutants were performed using SEM-EDS. Compared with the magnetic particles in atmospheric dustfall (Fig. 10a–d), those from vehicle exhausts consist of only three types of particles, including angular shapes with coarse surface textures, spherules and aggregates (Fig. 10e–g), while all magnetic particle types in dustfall samples were identified in the fly ashes (Fig. 10h–k). The EDS analysis showed that the major elements of the same three types of magnetic particles in vehicle exhausts and fly ashes are Fe and O, consistent with elemental features of those in atmospheric dustfall (Fig. 10l–n). This suggests that vehicle exhausts and fly ashes are the main pollutant sources of the dustfall. However, there are some differences in the compositions of the minor elements in the three types of particles between vehicle exhausts and fly ashes. Angular particles with coarse surface textures from vehicle exhausts contain more S, Cr, Cu, Zn and Mn, while those from

fly ashes have more Ca and Mn. Aggregates consist of more Cr, Zn and S in vehicle exhausts, whereas Ca and S are enriched in fly ashes. Spherules from vehicle exhausts contain higher amounts of heavy metals (Cr, Ni Mn and Zn), while those from fly ashes have higher Ca and Mn. Coarse-grained porous magnetic particles are only observed in fly ashes, characterized by a relatively low Fe concentration and high crustal elements (e.g. Si, Al, K, Ca, Mg, and Ti).

The EDS elemental data clearly indicate that the magnetic particles from vehicle exhausts contain higher concentrations of a greater range of elements from anthropogenic activities (S, Cr, Cu, Zn, Ni and Mn) than those from fly ashes, whose EDS spectra show a substantial peak of Ca. The $\chi_{lf}$ values of vehicle exhausts ($925.7 \times 10^{-8}$ m$^3$ kg$^{-1}$) and the Bell Tower street dust ($618.7 \times 10^{-8}$ m$^3$ kg$^{-1}$) are significantly higher than those of the fly ashes ($769.9 \times 10^{-8}$ m$^3$ kg$^{-1}$) and the nearby dustfall ($537.9 \times 10^{-8}$ m$^3$ kg$^{-1}$), indicating a higher content of magnetic contaminants. In summary, the magnetic particles emitted by vehicle exhausts and thermal power plants can be distinguished by a combination of morphological and elemental characteristics, which indicates that SEM-EDS can be used to trace the sources of anthropogenic pollutants in Xi'an.

## 5 Conclusions

By comparing the magnetic properties of surface sediments in natural dust sources in East Asia and various urban dust samples in Xi'an, we found that distal natural dust and local anthropogenic dust have different magnetic (i.e. $\chi_{lf}$ and $\chi_{fd}$), morphological and elemental characteristics. We take natural surface sediments as representative of the distal and natural dust, and urban street dust and anthropogenic pollutants as representatives of local and anthropogenic dust. Based on this end-member configuration, relative contributions of local and anthropogenic sources to urban atmospheric dustfall can be quantitatively estimated. The results show that both local and anthropogenic contributions decrease in spring and increase in other seasons, the opposite to seasonal variations of the dust flux.

SEM-EDS analysis of urban dust indicates that magnetic particles produced by anthropogenic activities have distinct morphological and elemental characteristics. The anthropogenic particles exhibit angular, spherical, aggregate, and porous shapes, and contain distinctive marker elements such as S, Cr, Cu, Zn, Ni, Mn and Ca. Based on these morphological and elemental characteristics, the porous particles are likely derived from emissions from the thermal power plant, while others





may be attributed to both vehicle exhausts and emissions from the thermal power plant. Our results suggest that magnetic signatures combined with morphological and elemental compositions can be used to quantitatively estimate the local and anthropogenic contributions to urban dust aerosols.

Acknowledgements. We thank Min Zhao and Hua Wang for the help in sample collection. We are also grateful for the help of Maojie Yang with SEM-EDS measurements, and Jan Bloemendal with language polishing. This work was supported by the National Key Research and Development Program of China (2016YFA0601902) and the Open Foundation of State Key Laboratory of Loess and Quaternary Geology (SKLLQG1631).

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



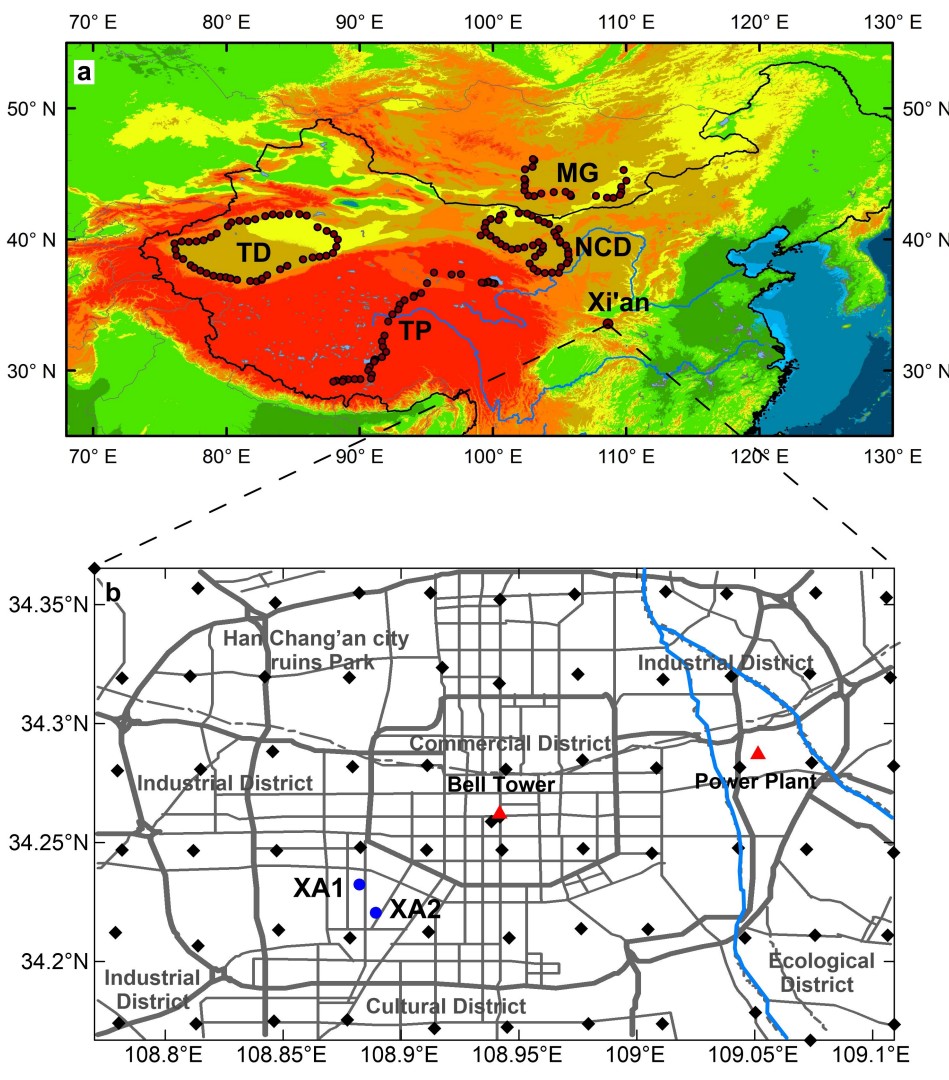

Figure 1. Locations of natural dust samples in the East Asian sources (a) and urban dust samples in Xi'an (b). NCD - northern Chinese deserts, MG - Mongolian Gobi, TD - Taklimakan Desert, and TP - Tibetan Plateau. Black diamonds are street dust sampling sites; blue dots are samples of consecutive atmospheric dustfall (XA1 at the Institute of Earth Environment, Chinese Academy of Sciences; XA2 at the Xinxinjiayuan residential community); red triangles are typical heavily-polluted sites, including the Bell Tower in an area of high traffic density, and the Baqiao thermal power plant.



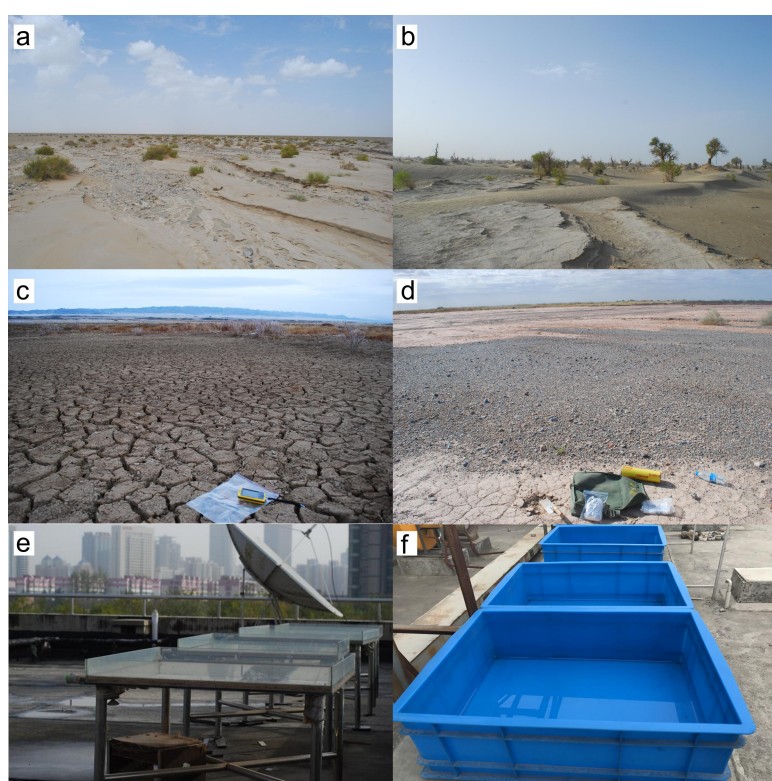

Figure 2. Sampling sites of surface sediments in a dry riverbed (a), desert margin (b), drainage depressions within sandy desert (c), and Gobi deserts (d), and atmospheric dustfall at XA1 (e) and XA2 (f).



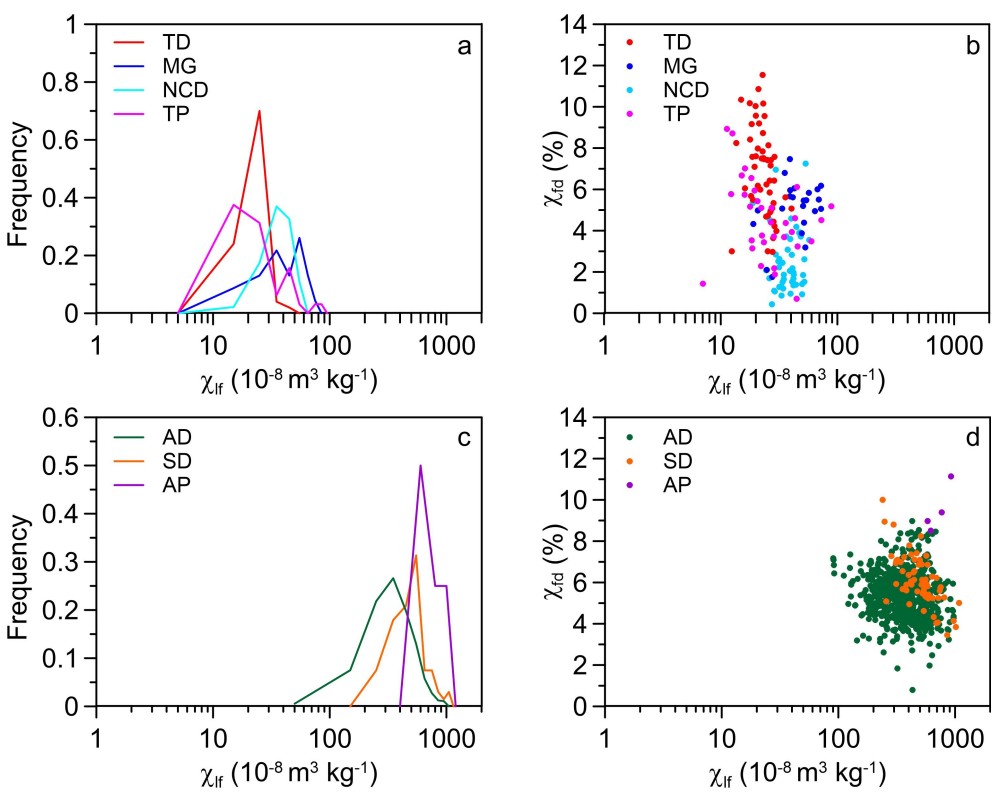

Figure 3. Frequency distribution of $\chi_{lf}$ and bivariate plots of $\chi_{lf}$ and $\chi_{fd}$ of natural surface sediments in each source (a, b) and urban dust aerosols (c, d), including atmospheric dustfall (AD), street dust (SD) and atmospheric pollutants (AP). Frequency distribution statistics of $\chi_{lf}$ for natural surface sediments, atmospheric dustfall and street dust, and atmospheric pollutants were generated using intervals of $10 \times 10^{-8}$ $m^3$ $kg^{-1}$, $100 \times 10^{-8}$ $m^3$ $kg^{-1}$ and $200 \times 10^{-8}$ $m^3$ $kg^{-1}$ respectively.



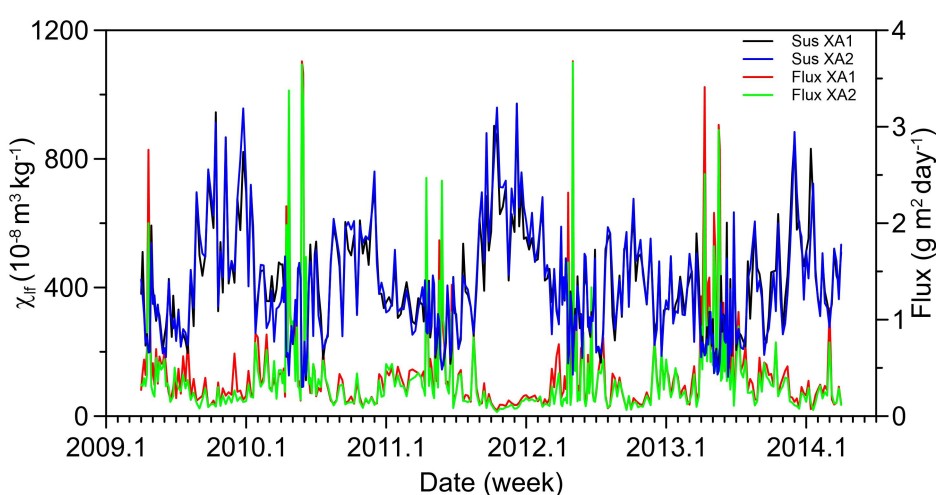

Figure 4. Time series of magnetic susceptibility and dust flux of atmospheric dustfall at XA1 and XA2, from 2009 to 2014.





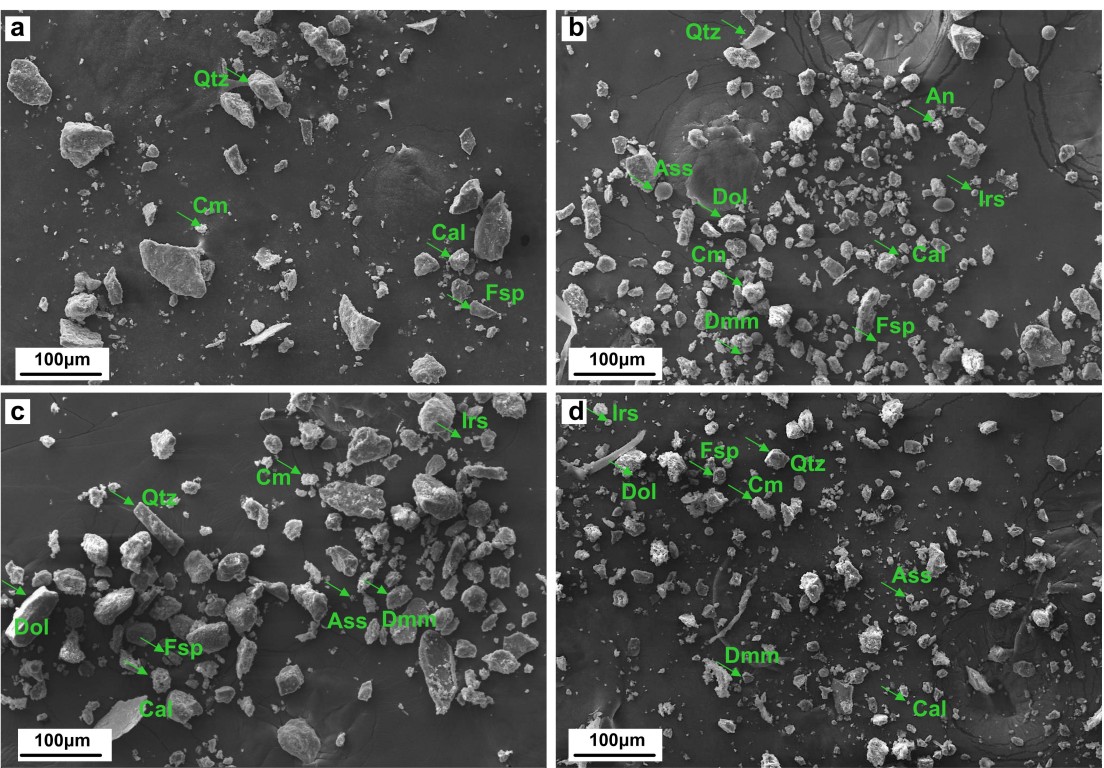

Figure 5. Morphology and mineralogy of representative samples of the natural surface sediments (a), street dust (b), atmospheric dustfall with low $\chi_{lf}$ (c) and high $\chi_{lf}$ (d). Qtz - quartz, Fsp - feldspar, Cal - calcite, Dol - dolomite, Cm - clay minerals, Dmm - detrital magnetic mineral, Irs - iron-rich sphere, Ass - aluminosilicate sphere, An - anomalous particles with a poriferous and loose structure.





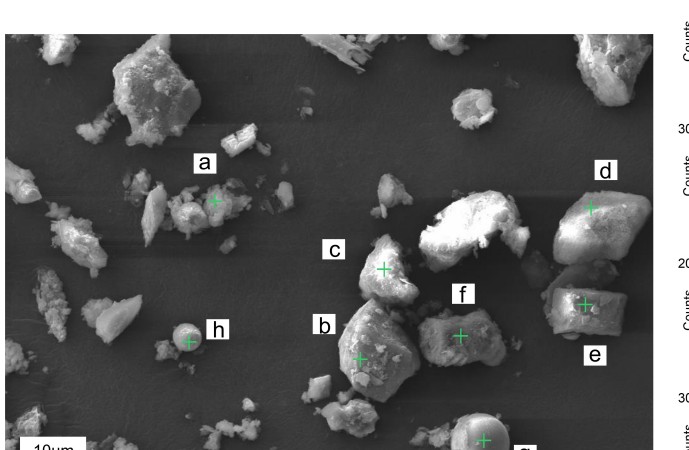

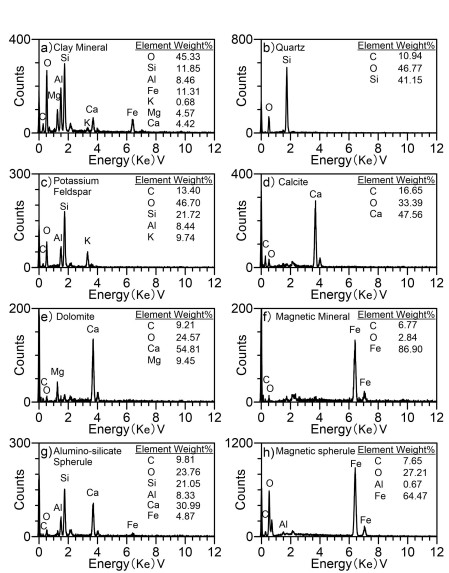

Figure 6. SEM photograph and elemental spectra for a typical sample of street dust. In the subplots the green plus symbols denote the locations of the beam used in the EDS analysis.



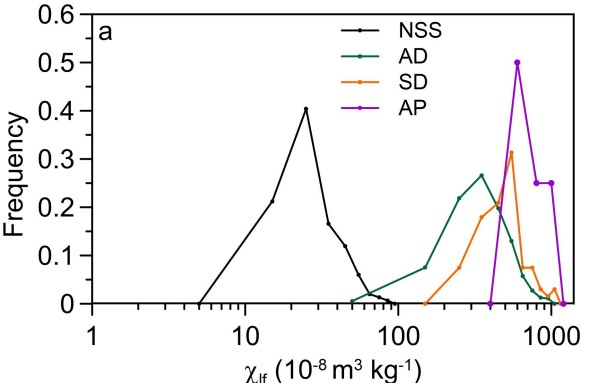 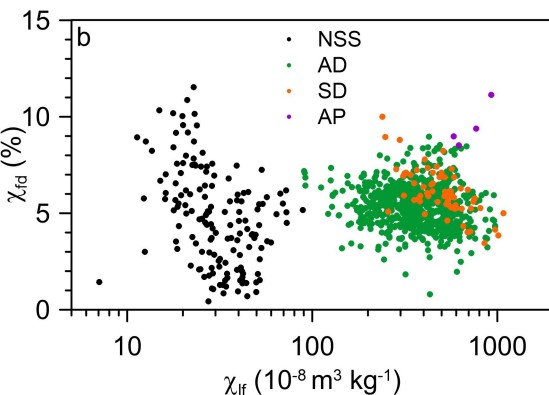

Figure 7. Frequency distributions of $\chi_{lf}$ (a) and bivariate-plots of $\chi_{lf}$ versus $\chi_{fd}$ (b) of natural surface sediments, street dust, atmospheric dustfall, and anthropogenic pollutants.



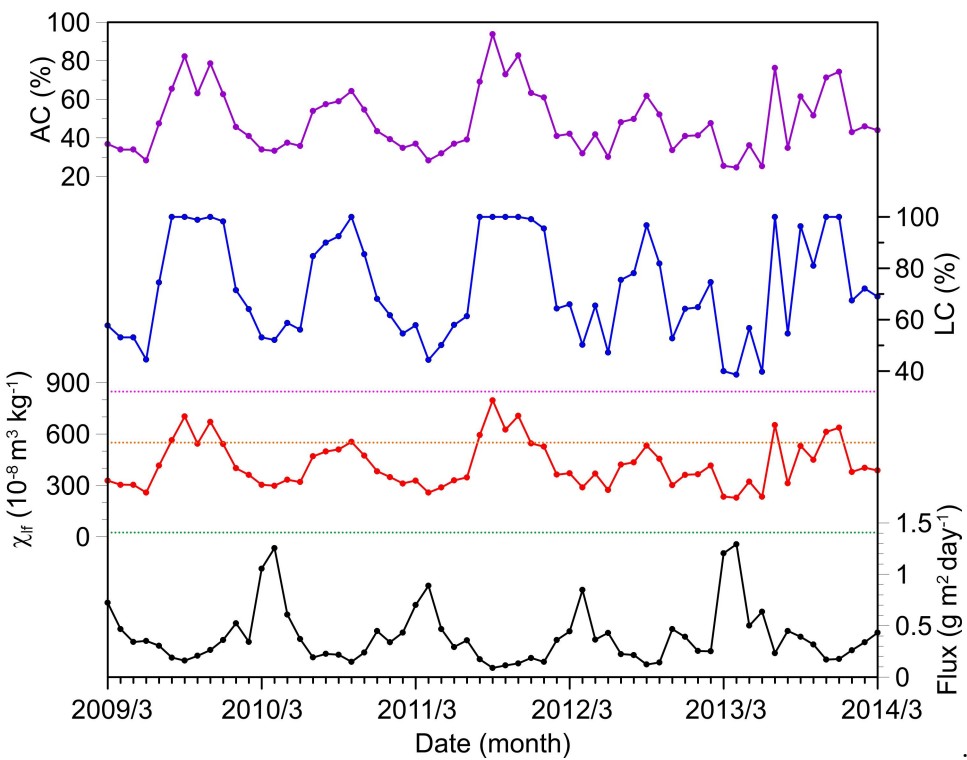

Figure 8. Monthly variations of dust flux (black) and $\chi_{lf}$ (red) of the atmospheric dustfall and estimated local and anthropogenic contributions (LC-blue, AC-violet). Green, orange and pink dotted lines denote the average $\chi_{lf}$ values of natural/distant dust, local dust, and polluted particles, respectively.




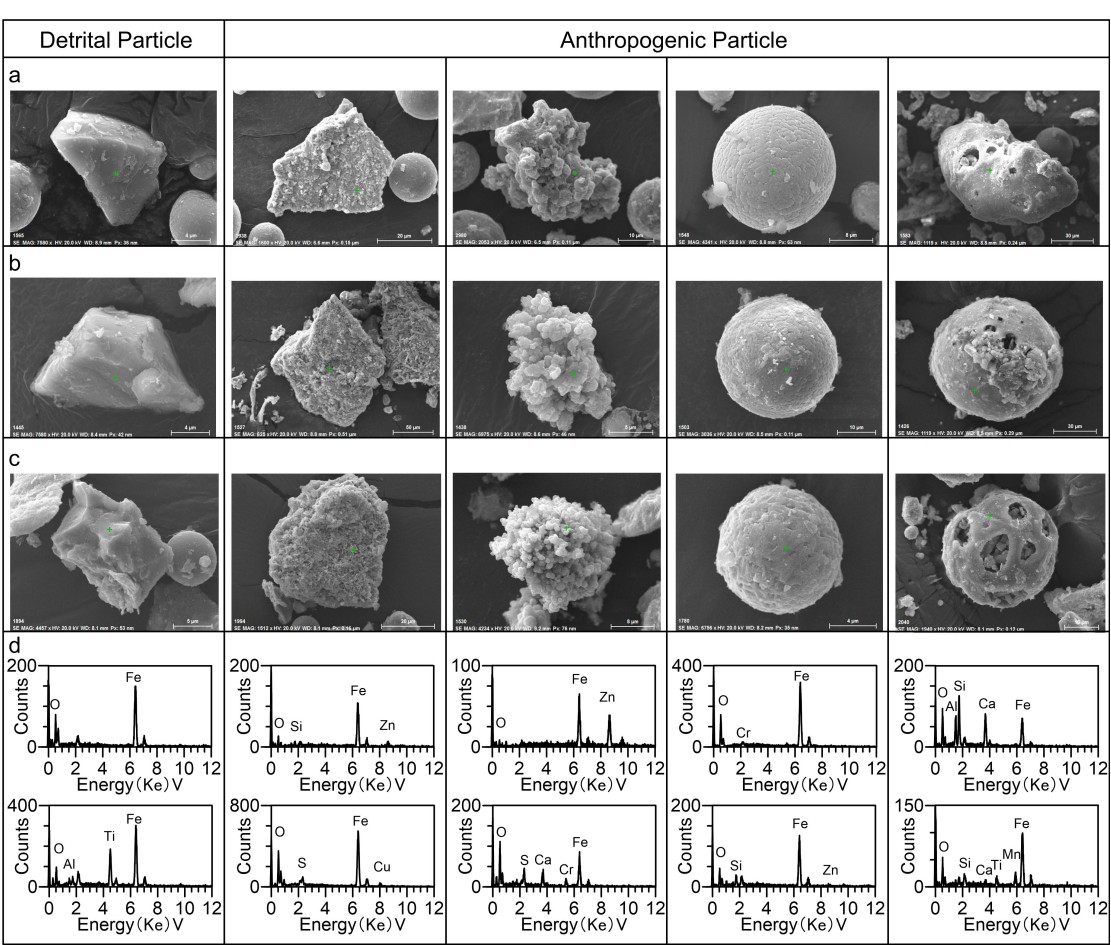

Figure 9. SEM images and typical elemental spectra (d) of magnetic extracts from the street dust (a), atmospheric dustfall with high $\chi_{lf}$ (b) and low $\chi_{lf}$ (c). From left to right, the particle morphologies represent detrital particles with relatively smooth surfaces from natural source regions, and anthropogenic particles with angular shapes and coarse surface textures, spherules, aggregates, and porous feature.



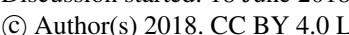

Figure 10. SEM images and elemental spectra of magnetic extracts from atmospheric dustfall (a–d), vehicle exhausts (e–g) and fly ashes (h–k) from the Baqiao thermal power plant. Black lines are elemental spectra of atmospheric dustfall. Blue and red lines are elemental spectra for vehicle exhausts and fly ashes.

