# Peer review of "Magnetic signatures of natural and anthropogenic sources of urban dust aerosol"

_Atmospheric Chemistry and Physics, 2018_

## Referee Comment (RC1) · Anonymous Referee #1 · 3 Jul 2018

Urban dust aerosols are important not only for pollution studies, but also for the global climatic changes. It is difficult to distinguish and quantify contributions from natural and anthropogenic origins. In this study, authors provided an excellent example from Xi'an, Northern China. The study includes two relevant parts: 1) characterization of these two kinds magnetic particles using X (Xfd) and SEM/EDS methods; and 2) a time series of dust flux between 2009 and 2014. The current data sets greatly improve our understanding of the natural and anthropogenic origins inputs to the Xi'an city. However, I think the data sets are not complete especially for ACP with such a high IF. Specifically, authors used magnetic methods to quantify the magnetic particles, but they only measured X and Xfd. X is affected by many factors, e.g., concentration, grain size, mineral types. The current study focused only on concentration without

information of grain size and mineral types. I strongly encourage authors to provide a more comprehensive study on this issue.

---

## Referee Comment (RC2) · Anonymous Referee #2 · 17 Jul 2018

This paper archives magnetic monitoring data for five years. In addition, detailed SEM/EDX data of anthropogenic particulates give a basic idea why the magnetic approach is useful to monitor the anthropogenic pollution especially by biomass burning.

Major points: 1. Authors mainly applied magnetic susceptibility to resolve the natural and anthropogenic signitures. Because Xlf and Xfd can be controlled by various factors including mineralogy and grain-size, more detailed magnetic data should greatly improve the quality of this paper. 2. As shown in Figures 4 and 8, Xlf shows an opposite trend to dust flux. This means that dominant anthropogenic magnetic signals were diluted by less magnetic natural dust input. Hence, the total quantitative anthropogenic matters are not varied but qualitative contribution is reduced as a result of increasing natural dust flux (Figure 8). Such a qualitative result may be estimated only by dust

flux ratio without Xlf data. To clarify the discussion 4.1, authors are highly encouraged to present quantitative magnetic data such as a saturation magnetization.

Minor points: 1. Figure 1: Insert a scale bar in a road-map. 2. Page 3, line 6: Check the reference (Maher et al., 1988) 3. Page 4, line 18: Sampling time? Fitering? Dust bag? Not enough information for sampling. 4. Page 6, lines 10 and 11: Why Xlf indicates different mineralogy? 5. Page 6, line 15: Difference in mean Xfd values of 6.9%, 5.1%, 4.6%, and 2.5% have any scientific meaning? 6. Page 7, line 13: Is that platinum or carbon coat for SEM observation?
* * *

---

## Author Comment (AC1) · 22 Sep 2018

Reviewer #1 Specifically, authors used magnetic methods to quantify the magnetic particles, but they only measured $\chi$ and $\chi$fd. $\chi$ is affected by many factors, e.g., concentration, grain size, mineral types.The current study focused only on concentration without information of grain size and mineral types. I strongly encourage authors to provide a more comprehensive study on this issue.

Reply: Thanks for this comment. We conducted more magnetic measurements (see Method section, Lines 11-24 in Page 5 in the revised version ). The temperature dependent susceptibility ($\chi$-T), hyteresis loops and first-order reversal curves (FORC) were used to better constrain the grain size and types of magnetic minerals (see Figs. 1 and

2, Lines 10-22 in Page 6 and Lines 1-5 in Page 7 in the revised version).

$\chi$-T is used to identify magnetic mineral composition. All the $\chi$-T heating curves (Fig. 1a-f) are characterized by a major susceptibility decrease at 580 degrees Celsius, i.e. the Curie temperature of magnetite, which pinpoint magnetite as the major contributor to $\chi$. All the samples are irreversible with cooling paths above heating trajectories due to the neoformation of magnetite (Jordanova et al., 2004; Kim et al., 2009). The $\chi$-T heating curves of the vehicle exhausts displays a decreasing $\chi$ between 580 and 700 degrees Celsius (Fig. 1b), suggesting the presence of hematite.

All samples have similar slightly wasp-waisted hysteresis loops (Fig. 1g-l). Their magnetic saturation was generally reached at a magnetic field of about 300 mT. This is a clear indication of the predominance of low coercivity ferrimagnetic minerals in all samples.

The Day plot and FORC diagram are powerful methods to identify the domain state distribution of magnetic materials (Day et al., 1977; Dunlop 2002a, b; Pike et al., 1999; Roberts et al., 2000; Harrison et al., 2008). All the samples agree well with single-domain (SD) + multi-domain (MD) admixture curves in the pseudo-single-domain (PSD) range of the Day plot (Fig. 2a).

The FORC diagrams for street dust (Fig. 2d), and anthropogenic pollutant (Fig. 2e) have divergent contours that are characteristic of MD grains. The FORC diagram for natural surface sediments seems to be characteristic of PSD/MD behavior, whose outer contours display divergent pattern and inner contours are somewhat less divergent (Fig. 2a). The FORC distributions of atmospheric dustfall (Fig. 2b) appear to have a mixed set of contours. The outer contours have a divergent pattern that would be expected for MD particles, while the inner contours close about a central peak represent SD grains.

Day, R., Fuller, M., Schmidt, V.A.: Hysteresis properties of titanomagnetites: grain-size and compositional dependence. Phys Earth Planet Inter., 13, 260–267,1977

Dunlop, D. J.: Theory and application of the Day plot (Mrs/Ms versus Hcr/Hc) 1. Theoretical curves and tests using titanomagnetite data. J. Geophys. Res., 107, 2076, https://doi.org/10.1029/2001jb000486, 2002a.

Dunlop, D.J.: Theory and application of the Day plot (Mrs/Ms versus Hcr/Hc) 2. Application to data for rocks,sediments, and soils. J. Geophys. Res ., 107, 2057, https://doi.org/10.1029/2001jb000487, 2002b.

Harrison, R. J., Feinberg, J. M.: FORCinel: An improved algorithm for calculating first- order reversal curve distributions using locally weighted regression smoothing, Geochem. Geophys. Geosyst., 9, Q05016, https://doi.org/10.1029/2008GC001987, 2008.

Jordanova, D., Hoffmann, V., and Febr, K. T.: Mineralmagnetic characterization of anthropogenic magnetic phases in the Danube river sediments (Bulgarian part), Earth planet. Sci. Lett., 30, 71–89, https://doi.org/10.1016/S0012-821X(04)00074- 3, 2004.

Kim, W., Doh, S. J., and Yu, Y. J.: Anthropogenic contribution of magnetic particulates in urban roadside dust, Atmos. Environ., 43, 3137-3144, https://doi.org/10.1016/j.atmosenv.2009.02.056, 2009.

Please also note the supplement to this comment:
https://www.atmos-chem-phys-discuss.net/acp-2018-452/acp-2018-452-AC1-supplement.pdf

[Figure]

**Fig. 1.** $\chi$-T heating (red line) and cooling (blue line) curves (a-f) and magnetic hysteresis loops (g-l) of representative samples.

[Figure]

[Figure]

**Fig. 2.** (a) Day-plot of the ratios Mrs/Ms and Bcr/Bc for representative samples from NSS, AD, STD, and AP; (b-e) FORC diagrams for representative samples of NSS, AD, STD, and AP.

---

## Author Comment (AC2) · 22 Sep 2018

Reviewer #2

Major points:

1. Authors mainly applied magnetic susceptibility to resolve the natural and anthropogenic signitures. Because $\chi$lf and $\chi$fd can be controlled by various factors including mineralogy and grain-size, more detailed magnetic data should greatly improve the quality of this paper.

Reply: Thanks for your suggestion. We added more magnetic measurements to assess the grain size and mineral type (see a detailed response to the first reviewer's comment).

[Figure]

2. Such a qualitative result may be estimated only by dustflux ratio without $\chi$lf data. To clarify the discussion 4.1, authors are highly encouraged to present quantitative magnetic data such as a saturation magnetization.

Reply: To clarify the discussion 4.1, we used the method of running median (Härdle and Steiger, 1995; Zhen and Yan, 1988; Marron, 1987; Mudelsee, 2006) to estimate the background of the observed weekly dust flux (see Fig. 2a, Lines 8-19 in Page 10 in the revised version) and then calculate monthly local anthropogenic contributions LCflux by ratio of monthly local background and total dust flux (see Fig. 2a, Lines 20-22 in Page 10 and lines 1-2 in Page 11 in the revised version).

Saturation magnetization (Ms) of representative samples (Fig.1 h-m) were measured to identify concentration of ferrimagnetic minerals. We found that the averaged values of Ms in different sources show a rising trend from the natural surface sediments (0.04 Am2/kg) to atmospheric dustfall (0.81 Am2/kg) and street dust (1.03 Am2/kg), and then to anthropogenic pollutant (1.58 Am2/kg), which correspond to the characteristics of averaged $\chi$lf in different sources. This indicates that the high $\chi$lf of urban dust is caused by the ferrimagnetic mineral from local anthropogenic source. In consequence, the LC contribution could also be estimated by the average $\chi$lf (25×10−8 m3 kg-1) of the surface sediments and local street dust (550×10-8 m3 kg-1). On this basis, we calculated the LC$\chi$ (see Fig. 2b, Lines 3-12 in 11 Page in the revised version).

The result showed that LCflux and LC$\chi$ values have the same trend and show a distinctive seasonal pattern (Fig. 2a-b), with the maximum in autumn (92.4 %, 92.3%), followed by winter (90.8 %, 74.7 %), summer (83.5 %, 71 %), and spring (73.0 %, 53.1%). Both the LCflux and LC$\chi$ are the lowest in spring, implying that distant natural dust input makes a great contribution to atmospheric dustfall during this period.

The LC variation exhibits a similar seasonal pattern with $\chi$lf, but opposite trend to that of dust flux (Fig. 2a-b). This means that dominant anthropogenic magnetic signals were diluted by less magnetic natural dust input. Hence, the local contribution is reduced as

a result of increasing natural dust flux in spring (see Fig. 2a-b, Lines13-21 in Page 11 in the revised version).

Minor points:

1. Figure 1: Insert a scale bar in a road-map.

Reply: We inserted a scale bar in the top left corner of road-map in Fig. 3b.

2. Page 3, line 6: Check the reference (Maher et al., 1988)

Reply: We checked the reference and deleted it.

3. Page 4, line 18: Sampling time? Fitering? Dust bag? Not enough information for sampling.

Reply: We added description on sampling time. The sample of fly ashes were taken from dust bag of electrostatic precipitators at the Baqiao thermal power plant (see Lines 16-21 in Page 4 in the revised version).

4.Page 6, lines 10 and 11: Why $\chi$lf indicates different mineralogy?

Reply: We corrected this sentence to "The different distribution patterns of $\chi$lf indicate that the assemblage of magnetic minerals in the NCD and TD may different from those in the MG and TP" (see Lines 18-19 in Page 7 in the revised version).

5. Page 6, line 15: Difference in mean $\chi$fd values of 6.9%, 5.1%, 4.6%, and 2.5% have any scientific meaning?

Reply: $\chi$fd is sensitive to the superparamagnetic (SP) component. There are virtually no SP grains when $\chi$fd is < 2 %, while a mixture of SP and coarser grains is indicated with $\chi$fd in the range of 2-10% (Dearing et al. 1994) (see Lines 9-11 in Page 7).

6. Page 7, line 13: Is that platinum or carbon coat for SEM observation?

Reply: Samples were mounted on SEM stub with the double-sided carbon tape and then coated with thin gold film (see Lines 4-5 in Page 6).

[Figure]

Dearing, J. A.: Environmental Magnetic Susceptibility, Chi Publishing, Kenilworth, UK, 1994.

Härdle, W., Steiger, W.: Algorithm AS 296: Optimal median smoothing, Journal of the Royal Statistical Society. Series C (Applied Statistics)., 44, 258-264, 1995.

Marron, J. S.: What does Optimal Bandwidth Selection Mean for Nonparametric Regression Estimation?, Department of Statistics, University of North Carolina at Chapel Hill, 1986.

Mudelsee, M.:Short note: CLIM-X-DETECT: A Fortran 90 program for robust detection of extremes against a time-dependent background in climate records, Computers & Geosciences., 32, 141-144, https://doi.org/10.1016/j.cageo.2005.05.010, 2006.

Zheng, Z. G., Yang, Y.:Cross-validation and median criterion, Statistica Sinica., 8 , 907–921, 1998.

Please also note the supplement to this comment:
https://www.atmos-chem-phys-discuss.net/acp-2018-452/acp-2018-452-AC2-supplement.pdf

[revised manuscript text omitted]

---

## Author Response (AR1)

**Author's Response**

**Reviewer #1**

**Comment:** Specifically, authors used magnetic methods to quantify the magnetic particles, but they only measured $\chi$ and $\chi_{fd}$. $\chi$ is affected by many factors, e.g., concentration, grain size, mineral types. The current study focused only on concentration without information of grain size and mineral types. I strongly encourage authors to provide a more comprehensive study on this issue.

**Reply:** Thanks for this comment. We conducted more magnetic measurements (see Method section, Lines 11-23 in Page 5 in the revised version). The temperature dependent susceptibility ($\chi$-T), hyteresis loops and first-order reversal curves (FORC) were used to better constrain the grain size and types of magnetic minerals (see Figs. 3 and 4, Lines 10-22 in Page 6 and Lines 1-5 in Page 7 in the revised version).

$\chi$-T is used to identify magnetic mineral composition. All the $\chi$-T heating curves (Fig. 3a-f) are characterized by a major susceptibility decrease at 580 ℃, i.e. the Curie temperature of magnetite, which pinpoint magnetite as the major contributor to $\chi$. All the samples are irreversible with cooling paths above heating trajectories due to the neoformation of magnetite (Jordanova et al., 2004; Kim et al., 2009). The $\chi$-T heating curves of the vehicle exhausts displays a decreasing $\chi$ between 580 and 700 ℃ (Fig. 3b), suggesting the presence of hematite.

All samples have similar slightly wasp-waisted hysteresis loops (Fig. 3g-l). Their magnetic saturation was generally reached at a magnetic field of about 300 mT. This is a clear indication of the predominance of low coercivity ferrimagnetic minerals in all samples.

The Day plot and FORC diagram are powerful methods to identify the domain state distribution of magnetic materials (Day et al., 1977; Dunlop 2002a, b; Pike et al., 1999; Roberts et al., 2000; Harrison et al., 2008). All the samples agree well with single-domain (SD) + multi-domain (MD) admixture curves in the pseudo-single-domain (PSD) range of the Day plot (Fig. 4a). The FORC diagrams for street dust (Fig. 4d), and anthropogenic pollutant (Fig. 4e) have divergent contours that are characteristic of MD grains. The FORC diagram for natural surface sediments seems to be characteristic of PSD/MD behavior, whose outer contours display divergent pattern and inner contours are somewhat less divergent (Fig. 4a). The FORC distributions of atmospheric dustfall (Fig. 4b) appear to have a mixed set of contours. The outer contours have a divergent pattern that would be expected for MD particles, while the inner contours close about a central peak represent SD grains.

Day, R., Fuller, M., Schmidt, V.A.: Hysteresis properties of titanomagnetites: grain-size and compositional dependence. Phys Earth Planet Inter., 13, 260-267,1977.

5    Dunlop, D. J.: Theory and application of the Day plot (Mrs/Ms versus Hcr/Hc) 1. Theoretical curves and tests using titanomagnetite data, J. Geophys. Res., 107, 2076, https://doi.org/10.1029/2001jb000486, 2002a.

Dunlop, D. J.: Theory and application of the Day plot (Mrs/Ms versus Hcr/Hc) 2. Application to data for rocks, sediments, and soils, J. Geophys. Res., 107, 2057, https://doi.org/10.1029/2001jb000487, 2002b.

Harrison, R. J., Feinberg, J. M.: FORCinel: An improved algorithm for calculating first-order reversal curve distributions
10    using locally weighted regression smoothing, Geochem. Geophys. Geosyst., 9, Q05016, https://doi.org/10.1029/2008GC001987, 2008.

Jordanova, D., Hoffmann, V., and Febr, K. T.: Mineralmagnetic characterization of anthropogenic magnetic phases in the Danube river sediments (Bulgarian part), Earth planet. Sci. Lett., 30, 71-89, https://doi.org/10.1016/S0012-821X(04)00074-3, 2004.

15    Kim, W., Doh, S. J., and Yu, Y. J.: Anthropogenic contribution of magnetic particulates in urban roadside dust, Atmos. Environ., 43, 3137-3144, https://doi.org/10.1016/j.atmosenv.2009.02.056, 2009.

[Figure]

Fig 3. χ-T heating (red line) and cooling (blue line) curves (a-f) and magnetic hysteresis loops (g-l) of representative samples of NSS (MG0907), atmospheric dustfall (AD, 2013.7.18 and 2010.4.30), street dust (STD, L5-10) and anthropogenic pollutant (AP): fly ashes (DCYH ) and vehicle exhausts (QCWQ).

[Figure]

Fig. 4 (a) Day-plot of the ratios $M_{rs}/M_s$ and $B_{cr}/B_c$ for representative samples from NSS, AD, STD, and AP, grain size boundaries and the SD+MD matrix line are according to Dunlop (2002). Percentages in the Day plot represent the concentrations of MD in the SD+MD mixture; (b-e) FORC diagrams for representative samples of NSS, AD, STD, and AP.

**Reviewer #2**

**Major points:**

**1. Comment:** Authors mainly applied magnetic susceptibility to resolve the natural and anthropogenic signitures. Because $\chi_{lf}$ and $\chi_{fd}$ can be controlled by various factors including mineralogy and grain-size, more detailed magnetic data should greatly
5      improve the quality of this paper.

**Reply:** Thanks for your suggestion. We added more magnetic measurements to assess the grain size and mineral type (see a detailed response to the first reviewer's comment).

2.  **Comment:** Such a qualitative result may be estimated only by dustflux ratio without $\chi_{lf}$ data. To clarify the discussion 4.1,
10     authors are highly encouraged to present quantitative magnetic data such as a saturation magnetization.

**Reply:** To clarify the discussion 4.1, we used the method of running median (Härdle and Steiger, 1995; Zhen and Yan, 1988; Marron, 1987; Mudelsee, 2006) to estimate the background of the observed weekly dust flux (see Fig. 10a, Lines 8-18 in Page 10 in the revised version) and then calculate monthly local anthropogenic contributions $LC_{flux}$ by ratio of monthly local
15     background and total dust flux (see Fig. 10a, Lines 19-22 in Page 10 and lines 1-2 in Page 11 in the revised version).

*Saturation magnetization ($M_s$) of representative samples (Fig.3h-m) were measured to identify concentration of ferrimagnetic minerals. We found that the averaged values of $M_s$ in different sources show a rising trend from the natural surface sediments (0.04 $Am^2/kg$) to atmospheric dustfall (0.81 $Am^2/kg$) and street dust (1.03 $Am^2/kg$), and then to anthropogenic pollutant (1.58 $Am^2/kg$), which correspond to the characteristics of averaged $\chi_{lf}$ in different sources. This*
20     *indicates that the high $\chi_{lf}$ of urban dust is caused by the ferrimagnetic mineral from local anthropogenic source. In consequence, the LC contribution could also be estimated by the average $\chi_{lf}$ ($25 \times 10^{-8}$ $m^3$ $kg^{-1}$) of the surface sediments and local street dust ($550 \times 10^{-8}$ $m^3$ $kg^{-1}$). On this basis, we calculated the $LC_{\chi}$ (see Fig.10b, Lines 3-12 in 11 Page in the revised version).*

*The result showed that $LC_{flux}$ and $LC_{\chi}$ values have the same trend and show a distinctive seasonal pattern (Fig. 10a-b),*
25     *with the maximum in autumn (92.4 %, 92.3%), followed by winter (90.8 %, 74.7 %), summer (83.5 %, 71 %), and spring (73.0 %, 53.1%). Both the $LC_{flux}$ and $LC_{\chi}$ are the lowest in spring, implying that distant natural dust input makes a great contribution to atmospheric dustfall during this period (see Fig.10a-b, Lines13-16 in Page 11 in the revised version). The LC variation exhibits a similar seasonal pattern with $\chi_{lf}$, but opposite trend to that of dust flux (Fig. 10a-b). This means that dominant anthropogenic magnetic signals were diluted by less magnetic natural dust input. Hence, the local contribution is*
30     *reduced as a result of increasing natural dust flux in spring (see Fig.10a-b, Lines1-4 in Page 14 in the revised version)..*

**Minor points**

**1. Comment:** Figure 1: Insert a scale bar in a road-map.

**Reply:** We inserted a scale bar in the top left corner of road-map in Fig.1 (see Fig. 1b).

**2. Comment:** Page 3, line 6: Check the reference (Maher et al., 1988)

**Reply:** We checked the reference and deleted it.

**3. Comment:** Page 4, line 18: Sampling time? Fitering? Dust bag? Not enough information for sampling.

**Reply:** We added description on sampling time. The sample of fly ashes were taken from dust bag of electrostatic precipitators at the Baqiao thermal power plant (see Lines 16-21 in Page 4 in the revised version).

**4. Comment:** Page 6, lines 10 and 11: Why $\chi_{lf}$ indicates different mineralogy?

**Reply:** We corrected this sentence to "*The different distribution patterns of $\chi_{lf}$ indicate that the assemblage of magnetic minerals in the NCD and TD may different from those in the MG and TP*" (see Lines 18-19 in Page 7 in the revised version).

**5. Comment:** Page 6, line 15: Difference in mean $\chi_{fd}$ values of 6.9%, 5.1%, 4.6%, and 2.5% have any scientific meaning?

**Reply:** *$\chi_{fd}$ is sensitive to the superparamagnetic (SP) component. There are virtually no SP grains when $\chi_{fd}$ is < 2 %, while a mixture of SP and coarser grains is indicated with $\chi_{fd}$ in the range of 2-10% (Dearing et al. 1994)* (see Lines 9-11 in Page 7).

**6. Comment:** Page 7, line 13: Is that platinum or carbon coat for SEM observation?

[revised manuscript text omitted]

---

## Author Response (AR2)

**Author's Response**

**1. Comment:** Please do not abbreviate these locations in the text. It greatly harms the readability. The number of instances of these phrases is not so high as to make the manuscript problematically longer. If you want to use these abbreviations in figures and/or tables, that would be more acceptable.

10

Reply: All the unnecessary and improper abbreviations are modified.

**2. Comment:** I think it would be better if you presented all of the data in this section as a figure rather than numbers. Your goal is to compare values, which is very difficult to do from the text. Also, I suggest you use more than ranges to convince the reader that location A is different from location B. Ranges don't use all the data in an effective manner, and are subject to problems with outliers. You could report mean (or median), and 1 and 2 standard deviations of the measurements, for instance. Or percentiles of the distribution of values. I think pretty much anything is better than a pure range. In other words, supplement figure 5 with one or two more figures that lead the reader to the conclusion you desire. Remove most, if not all, of the numbers from this paragraph. It's a very ineffective mode of communication.

**Reply:** We rewrote the chapter 3.3 and chose the mean values and standard deviations to compare the differences between various sources (Line 7-24 in Page 7 and Line 1-6 in Page 8). According to your suggestion, we added two figures in Figure 5.

Figure 5. Frequency distribution of  $\chi_{lf}$  (a, c), bivariate plots of  $\chi_{lf}$  and  $\chi_{fd}$  (b, d) and average values and standard deviations of  $\chi_{lf}$  and  $\chi_{fd}$  (e, f) of NSS in each source and urban dust aerosols, including AD, STD and AP. Frequency distribution statistics of  $\chi_{lf}$  for NSS, AD and STD, and AP were generated using intervals of  $10 \times 10^{-8}$  m3 kg-1,  $100 \times 10^{-8}$  m3 kg-1 and  $200 \times 10^{-8}$  m3 kg-1, respectively.

5 **3.** Comment: This is a poor topic sentence for this section. First, mineralogy isn't determined from images, but on spectroscopy. Second, the title of the section is about morphology and mineralogy. You need to edit this beginning section

(Lines 17 in Page 8). Why does this matter? You should use whatever magnification necessary to generate the appropriate data (Lines 18 in Page 8).

**Reley:** We corrected the sentence to "SEM provides morphology information based on gray-scale intensity. The elemental composition is determined by the EDS detector. In order to compare the morphology and mineralogy characteristic of different dust sources, more than 40 fields of views of the representative bulk samples were randomly obtained for various types of particles." (Lines 8-10 in Page 8).

When we observed the particles, we used magnification large enough to make them clear. But we took photos of sample particles from different sources (Figure 7) at the same magnification to demonstrate the differences in their grain sizes and shapes.

**4. Comment:** Please estimate the uncertainty in the derived LC values (from both methods). For example, for LCX, the values of 25 and 550 are not exact, but subject to some uncertainty. What, then, does this mean for LCX? You should then plot reasonable uncertainty bounds on your figures (Figure 10).

**Reply:** We calculated the standard deviations of both methods as their uncertainty boundaries. The standard deviations of  $LC_{flux}$  and  $LC_x$  mean the fluctuation range of values and can be measured the degree of their dispersion effectively. We added the uncertainty boundaries to be added the uncertainty boundaries of the standard deviation in Figure 10.

15 the uncertainty bounds estimated by standard deviations in Figure 10.